# Meta-Analysis Identifies *BDNF* and Novel Common Genes Differently Altered in Cross-Species Models of Rett Syndrome

**DOI:** 10.3390/ijms231911125

**Published:** 2022-09-22

**Authors:** Florencia Haase, Rachna Singh, Brian Gloss, Patrick Tam, Wendy Gold

**Affiliations:** 1School of Medical Science, Faculty of Medicine and Health, The University of Sydney, Camperdown, NSW 2006, Australia; 2Kids Neuroscience Centre, Kids Research, Children’s Hospital at Westmead, Westmead, NSW 2145, Australia; 3Molecular Neurobiology Research Laboratory, Kids Research, Children’s Hospital at Westmead, Westmead, NSW 2145, Australia; 4School of Medicine Sydney, The University of Notre Dame, Chippendale, NSW 2007, Australia; 5Westmead Research Hub, Westmead Institute for Medical Research, Westmead, NSW 2145, Australia; 6Embryology Research Unit, Children’s Medical Research Institute, The University of Sydney, Camperdown, NSW 2006, Australia

**Keywords:** Rett syndrome, WGCNA, *MECP2*

## Abstract

Rett syndrome (RTT) is a rare disorder and one of the most abundant causes of intellectual disabilities in females. Single mutations in the gene coding for methyl-CpG-binding protein 2 (MeCP2) are responsible for the disorder. MeCP2 regulates gene expression as a transcriptional regulator as well as through epigenetic imprinting and chromatin condensation. Consequently, numerous biological pathways on multiple levels are influenced. However, the exact molecular pathways from genotype to phenotype are currently not fully elucidated. Treatment of RTT is purely symptomatic as no curative options for RTT have yet to reach the clinic. The paucity of this is mainly due to an incomplete understanding of the underlying pathophysiology of the disorder with no clinically useful common disease drivers, biomarkers, or therapeutic targets being identified. With the premise of identifying universal and robust disease drivers and therapeutic targets, here, we interrogated a range of RTT transcriptomic studies spanning different species, models, and *MECP2* mutations. A meta-analysis using RNA sequencing data from brains of RTT mouse models, human post-mortem brain tissue, and patient-derived induced pluripotent stem cell (iPSC) neurons was performed using weighted gene correlation network analysis (WGCNA). This study identified a module of genes common to all datasets with the following ten hub genes driving the expression: *ATRX*, *ADCY7*, *ADCY9*, *SOD1*, *CACNA1A*, *PLCG1*, *CCT5*, *RPS9*, *BDNF*, and *MECP2*. Here, we discuss the potential benefits of these genes as therapeutic targets.

## 1. Introduction

Rett syndrome (RTT) is one of the most common genetic causes of intellectual disabilities in females and affects one in 10,000 births [1]. RTT is an X-linked dominant disorder caused by mutations in the *MECP2* gene, which encodes the Methyl-CpG Binding Protein 2 (MeCP2) protein. The molecular pathogenesis of RTT remains poorly understood, with patients presenting with numerous complex disabilities, which are likely due to the pleiotropic molecular functions of MeCP2 and its ubiquitous expression.

At a cellular level, MeCP2 expression is critical for neuronal maturation and neuronal function. In hemizygous male patients, loss-of-function mutations in *MECP2* cause neonatal encephalopathy, which is usually fatal before the age of two, and in heterozygous female patients it results in a severe neurological phenotype [2,3]. MeCP2 is a pleiotropic protein mediating early events of neurodevelopment, including neurogenesis, migration, and patterning, where it predominantly acts as a methyl-DNA binding protein and controls transcriptional regulation [4,5]. The protein is highly conserved among mammals showing 95% protein sequence identity between human and mouse, and to a lesser extent, between humans and zebrafish (48%). However, the genomic structure and expression patterns of MeCP2 in zebrafish and mammals are similar, suggesting probable conserved functions [6]. MeCP2 contains two highly evolutionarily conserved domains, the methyl-CpG-binding domain (MBD) and the transcriptional repression domain (TRD), and acts as a transcriptional repressor and activator. Upon binding to CpG islands, MeCP2 forms an inhibitory transcription complex through interactions of the transcriptional repression domain with cofactors, including Sin3A and histone deacetylase 1. MeCP2 also acts as a transcription activator to regulate gene expression by either long-range chromatin remodeling or by regulating RNA splicing [7,8,9]. MeCP2-deficiency leads to an excitation/inhibition (E/I) imbalance in the brain and is recognized as the leading cellular and synaptic hallmark of the disorder resulting in stereotypic hand movements, impaired motor coordination, breathing irregularities, seizures, and learning/memory dysfunctions [10].

Mice harbouring mutations in the *Mecp2* gene represent one of the most clinically relevant models for RTT as they recapitulate many of the features observed in RTT patients, such as seizures and motor and cognitive dysfunction, which has assisted in our understanding of the underlying pathophysiology [11]. However, despite the vast majority of RTT patients being female, most gene therapy and other preclinical studies in animal models of RTT have used male mice, which is not truly representative of the patient population. 

Even though the phenotype of the RTT mouse models is very robust, there are many differences in brain development and structure between humans and mice that may confound findings in translational preclinical studies [12]. For example, the origin of cortical neurons in brain development differs in humans and mice with the subventricular zone, where human neurogenesis mostly occurs, that is significantly reduced in mice [13]. Thus, regardless of the significant insight gained from these models, inconsistencies between mouse models and human disease may affect the validity of preclinical findings.

Immortalised cell lines and post-mortem brain tissue have also been used extensively to study the pathophysiology of RTT. However, the use of post-mortem brain tissue is limited, as the tissue only reflects end-stage disease and cannot be used in live-cell testing studies such as electrophysiology and immortalised cell lines do not represent the complex organisation of the brain [8].

More recently, stem cells, including human embryonic stem cells (hECSs) and induced pluripotent stem cells (iPSCs), have come to play an important role in in vitro disease modelling. hECSs are generated from early-stage human embryos and have the potential to differentiate into various cell types, whereas iPSCs are derived from patients and can be differentiated into any cell type [14]. Reprogramming of somatic cells to iPSCs through the overexpression of transcription factors was demonstrated over a decade ago [15] and this technology has now strengthened the utility of stem-cell-based disease models [16]. Over the past few years, several studies have successfully generated iPSC lines from RTT fibroblasts and have differentiated these lines into neural progenitor cells (NPCs), neurons, and glial cells [17]. Stem-cell-based modelling has been demonstrated to be effective for RTT research, because iPSC lines can harbour pathogenic *MECP2* mutations and thus can demonstrate neuronal morphological defects, such as reduced dendritic branching, spine density, and smaller soma size [18,19]. Several studies have reported differentiated neuronal cells from RTT-iPSCs in two-dimensional (2D) cultures, with a smaller soma size compared to that of controls [14,20,21,22,23]. Additionally, the dysregulation in cellular maturation and morphological complexities in RTT-iPSC neurons have recapitulated the findings of mouse studies and in human post-mortem brain tissues [24].

The complexities of RTT at a clinical level and MeCP2 function have resulted in significant challenges for developing safe and effective therapies [25]. It is unclear whether novel therapies that have shown promising preclinical efficacy would effectively mitigate systemic manifestations of the disease when administered in the clinic. This is partly due to the lack of models that cover all aspects of the disease. Thus, well-characterised, disease-relevant models are critical to uncovering underlying molecular, cellular, and physiological intermediate phenotypes in the pathophysiology of RTT that may provide insights into potential therapies. Therefore, we hypothesise that by taking advantage of all existing models, both old and new (Figure 1), useful insights into the pathophysiology of RTT may be gleaned, which will drive the discovery of novel therapeutic targets. To do this, we conducted a meta-analysis of the transcriptomic data from three different RTT models: mouse brain, post-mortem human brain tissue, and iPSC-derived neurons. Weighted gene correlation network analysis (WGCNA) offers a powerful method to untangle novel disease pathways compared to approaches, such as differential gene expression. Thus, this study has utilised WGCNA to examine three previously published transcriptomic datasets of human post-mortem brain tissue, iPSC-derived neurons, and mouse brain samples. After identifying a consensus module between the three datasets, we analysed the genes in that module against another two datasets which could not be included in the WGCNA analysis, using differential gene expression.

## 2. Results

### 2.1. Data Pre-Processing and Identification of Common Genes

Publicly available genome-wide transcriptomic datasets of iPSC-derived neurons, post-mortem human brain tissue, and mouse brains were retrieved from the NCBI Gene Expression Omnibus database. These included: GSE75303 (post-mortem), GSE123753 (iPSC-derived neurons) [26], and GSE96684 (mouse brain) [27] (Table 1). The post-mortem and mouse datasets included RTT and wild-type samples, whereas the iPSC-derived neurons included RTT and isogenic controls. The post-mortem dataset included sequencing results from both the temporal and frontal cortex. The age of the patients ranged from 17 to 20 years, and all subjects were female, harbouring three different mutations: c.378-2A > G, c.763C > T and c.451G > T. The mouse samples were from the brain cortex, and all were mouse nomenclature *Mecp2* knockout males [18]. The iPSC-derived neurons were females harbouring a deletion between exons 3 and 4 of *MECP2*. All samples were included in this study.

All datasets were normalised and filtered prior to WGCNA analysis (Figure 2). Briefly, abnormal samples were first filtered through hierarchical clustering, where any missing data count was eliminated. The genes in the mouse dataset were homologated to the human genome, with only the common genes being included in the study. Overall, there were a total of 9864 genes included in this analysis (Figure 2). Specific details on the normalisation used for the three different datasets can be found in Section 4.2.

### 2.2. Weighted Gene Co-Expression Networks

Weighted gene co-expression networks were constructed based on the identified genes following the soft threshold analysis using all three datasets combined. An optimal soft-thresholding power is needed to calculate co-expression similarity. Hence, to assess the similarity between genes at the expression and network topology levels, we created a topological overlap matrix (TOM) which was achieved by calculating the adjacency and correlation matrices of the gene expression profile. As shown in Figure 3A in the scale free topology plot, power 8 was the lowest power where all three datasets reached a topology fit index of 0.9. Hence, it was chosen to produce the hierarchical clustering tree (dendrogram). Using the hierarchical average linkage clustering method in combination with the TOM, we proceeded to identify gene modules of each gene network. The dynamic tree cut algorithm highlighted all gene modules and each was identified by a colour (Figure 3B). Each tree branch constitutes a module, and each leaf in the branch is one gene.

### 2.3. Correlation between Modules and Clinical Traits 

The module–trait associations were analysed by correlating module–sample eigengenes with clinical traits to identify significant associations. The colours of all the modules were selected at random to distinguish between modules. Correlation coefficients were assigned to each module and the disease status trait (RTT vs. wild-type (WT)). Subsequently, only modules with a significant correlation to the disease trait (*p* < 0.05) were identified for all three datasets (Figure 4).

### 2.4. Different Brain Tissues/Models Generate Modules of Dysregulated Genes

Through the identification of modules with a significant correlation coefficient to the disease trait across all models and tissues, four modules were found to be significantly dysregulated across all datasets (*p* < 0.5 eigenscore associated with disease trait): brown4, blue, magenta, and skyblue (Figure 4).

### 2.5. Module Analysis

To better understand the biological functions of the genes in the four modules, each module was subjected to KEGG pathway enrichment analysis (Figure 5). The level of significance of each pathway enrichment was calculated and expressed in adjusted *p*-values using the Bonferroni correction method. We then focussed on those pathways that had higher adjusted *p*-values (depicted in yellow in Figure 5). 

The brown4 module was highly enriched in cytokine–cytokine receptor interaction, the TGF-beta signalling pathway, fluid shear stress, and atherosclerosis, and signalling pathways regulating pluripotency of stem cells. The blue module was highly enriched in pathways including ribosomes, COVID-19 disease, thermogenesis, and basal transcription factors. The magenta module was enriched in cysteines and methionine metabolism, biosynthesis of amino acids, biosynthesis of cofactors, glycosaminoglycan degradation, mucin-type 0-glycan biosynthesis, pantothenate and CoA biosynthesis, and 2-oxocarboxylic acid metabolism. Finally, the skyblue module was enriched in GABAergic synapses, morphine addiction, vibrio cholerae infection, thyroid hormone synthesis, purine metabolism, cortisol synthesis and secretion, alcoholism, ovarian steroidogenesis, glutamatergic synapsis, cholinergic synapse, the longevity regulating pathway, growth hormone synthesis, secretion, and action, inflammatory mediator regulation of transient receptor potential (TRP) channels and relaxin signalling.

### 2.6. Key Cellular Pathways Involved in Synapses Dysregulated in Rett Models

Given the relevance to the known pathophysiology of RTT of the pathways identified in the skyblue module, we investigated this module further. Furthermore, focussing on the disease trait (WT vs. RTT), the skyblue module exhibited the highest correlation (Figure 4) and more disease-relevant enrichment (Figure 5). Therefore, this module was identified as a key module in RTT and was subjected to further analysis. 

Interestingly, we found that the skyblue module was driven by hub genes: *MECP2*, *BDNF*, *SOD1*, *PLCG1*, *CCT5*, *RPS9*, *ADCY9*, *ADCY7*, *ATRX,* and *CACNA1A* (Figure 6 and Table 2). Hub genes are defined as genes with connectivity (degree) greater than 10 in the genetic interaction network. All genes were shown to be downregulated in RTT except for *CCT5* and *ADCY9*.

### 2.7. Differential Gene Expression in iPSC Derived Neurons (OH) and Postmortem (MT) Datasets

To broaden the study and determine whether the skyblue module genes were also dysregulated in other RTT studies, the expression profiles obtained from two other datasets were analysed. The MT dataset (GSE6955 [28]) consisted of post-mortem human brain tissue, and OH (GSE107399 [29]) of iPSC-derived neurons (Table 3). Due to the constrains of WGCNA analysis, these datasets could not be included in the original analysis as the OH dataset had less than six samples (considering each experimental replicate as one sample) and the MT dataset was analysed using single cell RNA sequencing. The MT dataset consisted of six post-mortem superior frontal gyri samples, two belonging to female RTT patients, and four non-RTT controls (Table 3). The OH dataset consisted of seven iPSCs-derived neuron samples, of which two were RTT patients analysed in replicate, and three were the corresponding isogenic controls.

After performing differential gene expression analysis, 12,625 genes were identified in the MT dataset, of which 156 were significantly upregulated and 61 significantly downregulated (*p* < 0.05). Conversely, analysis of the OH data demonstrated 20,055 differentially expressed genes. Of these, 655 were significantly upregulated and 992 were significantly downregulated (*p* < 0.05).

Next, the gene expression profiles of MT and OH were cross-referenced with those genes in the skyblue module to identify common dysregulated genes across all datasets. Overall, there were 71 genes shared between the skyblue module and the MT dataset. *TOX3*, *FABP7*, *ATRX*, and *SGMS1* had the largest positive log-fold changes (Figure 7A). *BDNF*, *GNG11*, *FAM168B*, *HMOX1*, and *VCL* were identified as having the largest negative log-fold changes (Figure 7A). A comparable number of genes (107) were common between the skyblue module and the OH dataset. The top five upregulated genes with the greatest positive log-fold change were *NPAS4*, *FABP7*, *HECW2*, *TOX3*, and *CACNA1A* (Figure 7B). Conversely, *FREM2*, *BDNF*, *HMOX1*, *KDELR2*, and *CXADR* had the highest negative log-fold changes (Figure 7B).

The expression of the hub genes identified in the skyblue module was also cross-examined in the MT and OH datasets (Figure 8). From the meta-analysis, *CCT5* and *ADCY7* were upregulated in the skyblue module, whilst the remaining eight (*MECP2*, *BDNF*, *CACNA1A*, *ADCY9*, *ATRX*, *RPS9*, *SOD1*, and *PLCG1*) were downregulated. 

When compared with the other datasets, *ATRX* was significantly upregulated (*p* < 0.05) in the MT dataset and *CACN1A* was upregulated in the OH dataset. Interestingly, *BDNF* was significantly downregulated in all datasets (Figure 8A,B).

## 3. Discussion

The overarching aim of this study was to identify common pathways and genes that intersect RTT transcriptomic studies spanning different species and models with the premise of identifying universal and robust disease drivers and therapeutic targets. To do so, a meta-analysis and bioinformatics approach consisting of the identification of gene modules rather than differential gene expression was employed to interrogate the transcriptomic landscape of RTT using human post-mortem brain tissue, mouse models, and patient-derived neurons. 

After identifying the statistically significant dysregulated modules between all the RTT samples and controls, the module with the highest correlation to disease status and genes with the highest connectivity within the module were interrogated to identify key genetic drivers across all tissue samples and models. Reassuringly, the identified hub genes included *MECP2* and *BDNF*, where the correlation between the two genes in RTT is well recognized, with *BNDF* being a well-established target gene of MeCP2 [30,31].

### 3.1. Meta-Analyses Produced Four Significant Modules Correlated to Disease Status

Through this meta-analysis four modules of genes that were significantly dysregulated in the RTT transcriptome relative to the controls was identified. The pathways that were enriched in each of the four modules were investigated and the brown4 module was identified to be mostly enriched in pathways related to immunological aberrations, which is consistent with previously published studies in RTT, including one of our own [16,25,26]. Next, the magenta module was enriched for pathways primarily involving the metabolic system, which also aligns with previously reported literature [32,33]. On the other hand, enrichment of the blue module did not produce any pathways known to be of relevance in RTT. The fourth module, skyblue, consisted of enriched modules, including glutamatergic, GABAergic, and cholinergic synaptic pathways, as well as protein export, and was identified to have the most enriched pathways relevant to the neuropathology of RTT, hence supporting our further focus on this module.

### 3.2. Meta-Analysis Hub Genes within the Skyblue Module Are Relevant to RTT Pathology

The ten hub genes that were identified as the main drivers of the skyblue module were *ATRX*, *ADCY7*, *ADCY9*, *SOD1*, *CACNA1A*, *PLCG1*, *CCT5*, *RPS9*, *BDNF*, and *MECP2*. They are therefore surmised to play key roles in the pathology of RTT and may assist in understanding the underlying disease pathophysiology, as well as identifying disease drivers and drug targets. 

ATRX (ATRX Chromatin Remodeler) has recently been implicated in RTT as a binding partner of MeCP2 where together they modulate pericentric heterochromatin (PCH) organization in neurons [34]. Mutations in ATRX cause ATR-X syndrome, implicated in abnormal brain development and associated with severe intellectual disability [34,35]. The downregulation of ATRX in this meta-analysis supports previous reports of an interaction with MeCP2, where MeCP2 recruits the helicase domain of ATRX to heterochromatic foci in a DNA methylation dependent manner, as shown in living mouse cells [36]. Furthermore, it has been shown that the heterochromatin location of ATRX is disrupted in Mecp2-null mice neurons. These data together suggest that a MeCP2–ATRX interaction leads to pathological changes that contribute to the mental retardation phenotype. Interestingly, as an epigenetic modifier, ATRX has been implicated in cancer and has received a level of attention in the identification of expression modifying drugs [37]. ATRX loss leads to increased DNA damage and general genomic instability [38], and thus drugs or small molecules aimed at increasing the stability of the genome may be potential therapeutic options for RTT. 

Adenylate Cyclases 7 and 9 (*ADCY7* and *ADCY9*) are membrane-bound enzymes that catalyze the formation of cyclic AMP from ATP and are highly expressed in the brain. De novo mutations in *ADCY7* have been reported in autism spectrum disorders (ASD) where the gene has been proposed to be a risk factor [39]. ASD and RTT share some commonalities with RTT individuals showing some ASD-like behaviors [40,41]. *ADCY7* mRNA is highly expressed in microglia and plays an important role in presynaptic GABA release, and evidence suggests that *ADCY7* is involved in mood regulation and plays an essential role in the immune response [42]. Conversely, despite *ADCY9* being highly expressed in the brain, its function in the CNS remains largely unknown. However, some findings have suggested that *ADCY9* may regulate cognitive function and learning and memory [42]. Interestingly, *ADCY9* has been shown to be downregulated in Mecp2 null embryonic cortexes, suggesting *ADCY9* as a target of MeCP2 [43]. This effect is lost postnatally, suggesting the crucial role of *ADCY9* in embryogenesis [36,43]. Interestingly, both genes are involved in two common pathways: the *GPER1* signaling and integrin pathway, which provides potential therapeutic targets to explore in RTT. 

*SOD1* plays a crucial role in the oxidative stress response and systemic redox alterations, and the related oxidative stress is well reported in RTT [44]. It is therefore not surprising to find the free radical scavenger *SOD1* enzyme downregulated in this meta-analysis. Loss of *SOD1* has been hypothesized to result in an accumulation of mitochondrial reactive oxygen species, leading to oxidative damage and mitochondrial dysfunction [45]. Animal studies have suggested a possible direct correlation between *Mecp2* mutations and increased ROS levels, and the debate continues regarding whether oxidative stress is a cause or consequence of RTT. 

The voltage-dependent P/Q-type calcium channel subunit alpha-1A (*CACNA1A*) gene has been implicated in epileptic encephalopathy, familial hemiplegic migraine, episodic ataxia, and spinocerebellar ataxia [46] and has recently been reported in a small number of atypical Rett patients previously lacking known genetic mutations [47]. Voltage-sensitive calcium channels mediate the entry of calcium ions into excitatory neurons and are also involved in a variety of calcium-dependent processes and neurotransmitter release. Our findings suggest that the downregulation of *CACNA1A* in this meta-analysis may be contributing to the epileptic encephalopathy of RTT. Among its related pathways are the CREB and integrin signaling pathways. 

*PLCG1* (Phospholipase C, gamma 1) is a protein involved in cell growth, migration, apoptosis, and proliferation. Among its related pathways is theBDNF-TrkB signaling pathway. Even though no direct link to *MECP2* has been reported in the literature, it is known that activation of the neurotrophin receptor *TRKB* by *BDNF* triggers downstream *PLCG1* signaling [48].

No direct relation has been reported between *CCT5* and *RPS9* and *MECP2*. However, CCT5 is implicated in the cellular pathways related to trafficking to the periciliary membrane and cell cycle and has also been linked to intellectual disabilities and early onset motor neuropathies [49,50,51]. On the other hand, *RPS9* is linked to RNA binding and is a structural constituent of the ribosome, and as ribosomal dysfunction has been previously reported in RTT iNeurons by Rodrigues et al., 2020 [26], the dysregulation of *RPS9* in this study supports these findings and provides further evidence of ribosomal dysfunction in RTT. 

In addition, common cellular pathways, such as the CREB and integrin signaling pathways, are common amongst the hub genes. The CREB pathway has previously been reported to be implicated in RTT, where overexpression of CREB signaling in RTT forebrain neurons rescued the phenotype of neurite growth, dendritic complexity, and mitochondrial function [52]. Furthermore, pharmacological activation of CREB in female RTT mice rescued several behavioral phenotypes [52]. These findings support the motion to investigate the CREB pathway as a potential therapeutic target [52]. In addition, while the integrin pathway has not been reported in RTT, it has been previously implicated in dendritic development, autism spectrum disorder, and intellectual disabilities [53] suggesting that this pathway too could also be a potential target for future RTT therapeutics.

### 3.3. Meta-Analysis Shows Commonly Dysregulated Synaptic Pathways

Through this study, three synaptic pathways enriched in the skyblue module were identified, namely the cholinergic, glutamatergic, and GABAergic pathways. A loss of excitation/inhibition (E/I) balance in the neural circuit is a major hallmark of RTT pathology, causing many neurological symptoms, such as loss of purposeful hand movements, impaired motor coordination, breathing irregularities, and seizures, amongst others [10]. This loss of E/I balance is caused by MeCP2 deficiency, leading to a dysregulation of the glutamatergic and GABAergic pathways. Furthermore, downstream genes affected in RTT such as *BDNF* play an important role influencing neurotransmission activity. Many drugs have been tested to improve the E/I balance in RTT, including glutamatergic modulators such as AMPAkines to increase excitatory synapsis and enhance *BDNF* expression, ketamine, and NMDAR antagonist to enhance neuronal activity [54,55]. GABAergic modulators have also shown potential in aiding with behavioral dysfunction in RTT patients and mice. However, while respiratory alterations were ameliorated by treatment using benzodiazepines and Midazolan in mice, the phenotype was not fully rescued [56].

### 3.4. Expression of Overlapping Genes in MT and OH in Comparison to Skyblue

A comparison of differentially expressed genes in the MT and the OH datasets with the skyBlue module identified 71 and 107 commonly expressed genes, respectively. Interestingly, *TOX3* was upregulated in both datasets and the skyblue module. *TOX3* plays a role in shaping DNA and altering chromatin structure and while the protein has been shown to be a neuron survival factor [57], it is yet to be linked with neurodevelopmental disorders and specifically to RTT. *BDNF* and *HMOX1* were also commonly dysregulated in all the datasets where they were observed to be significantly downregulated. 

*HMOX1* is a heme oxygenase responsible for the degradation of heme to biliverdin/bilirubin and free iron and heavily implicated in aging and disease. The expression of HMOX1 is confined to small populations of neurons and glia and is upregulated by a wide range of pro-oxidant and other stressors [58]. While there have been no reports linking *HMOX1* to RTT pathology, its downregulation confirms the role of oxidative stress in the pathology of RTT. *ATRX* and *CACNA1A* were identified to be dysregulated in the skyblue module as well in either the MT or OH datasets where the expression of *ATRX* was upregulated in the MT dataset and *CACNA1A* was upregulated in OH. 

Additionally, BDNF, an identified hub gene for skyblue, was also downregulated in both the MT and OH datasets. This is the first time *BDNF* has been demonstrated to be consistently downregulated in a bioinformatic meta-analysis examining dysregulated genes across species and models.

### 3.5. Hub Gene Expression Comparison across Studies

Of the ten identified hub genes in the meta-analysis, eight were downregulated, suggesting that wild type MeCP2 transcriptionally activates these genes, and two (*ADCY9* and *CCT5*) were upregulated, suggesting that MeCP2 transcriptionally represses these genes. 

From the differential gene expression analysis performed on the OH and MT datasets, we showed that *BDNF* was downregulated in both studies, and *ARTX* and *CACN1A* were upregulated in the MT and OH datasets, respectively. We showed an overall trend of upregulation in the five tested genes *ATRX*, *ADCY7*, *ADCY9*, *CACNA1A*, and *SOD1.* These results were different to that found in the meta-analysis as only *ADCY9* was upregulated. This disparity in expression between the meta-analysis and differential gene expression points to the complexity of RTT and the context dependent expression of *MECP2*.

Here, we have used two different analytical tools (WGCNA and DGE), two species, and three models to identify dysregulated genes that drive the disease pathology of RTT. The identification of *BDNF* as the only consistent gene to be downregulated relative to controls across all models comes as no surprise given the known association with RTT. *BDNF* has been explored as a therapeutic target for RTT. However, as *BDNF* has a low blood–brain barrier permeability, this limits its bioavailability for peripheral administration as a therapy [59]. Three clinical trials aimed at augmenting *BDNF* expression, trialing Copaxone (glatiramer acetate) [60] and Fingolimod, have been conducted [61]. However, to date, no therapies have entered the clinic, with the glatiramer acetate trial being withdrawn due to reported potential life-threatening reactions [62]. Additional compounds have been described to increase *BDNF* levels and improve RTT-like symptoms in mice, however, none have reached human clinical trials, alluding to the complexity of the disorder and difficulty of this approach [59].

## 4. Materials and Methods

### 4.1. Dataset Selection 

The three datasets included in the WGCNA analysis were obtained from the NCBI Gene Expression Omnibus (GEO; https://www.ncbi.nlm.nih.gov/geo/ (last accessed on 20 June 2022)): GSE75303 (Post-mortem), GSE123753 (iPSC-derived neurons), and GSE96684 (Mouse Brain). The GSE75303 dataset contained 12 samples in total, including three female RTT patient frontal and temporal cortexes harboring mutations at c.378-2A > G, c.763C > T and c.451G > T and three female age-matched controls. The GSE123753 dataset consisted of six female samples: three patients involving rearrangements that removed exons 3 and 4, creating a functionally null mutation, and their three corresponding isogenic controls. The GSE96684 dataset consisted of eight male mouse samples: four MECP2 knockout and four wild type mice. The characteristics of the samples in each dataset are summarized in Table 1.

### 4.2. Dataset Pre-Processing

Since the three datasets were from different sequencing platforms, we performed pre-processing according to a previously published WGCNA pipeline. Specifically, GSE75303 (human post-mortem brain) array data were quantile normalized, GSE123753 (iPSCs derived neurons) provided the already quantile normalized data as a supplementary file through GEO (Gene Expression Omnibus), and the raw sequencing data were obtained for GSE96684 from SRA (PRJNA3779366) and mapped to mm10 using STAR (Version 2.7). Mouse gene symbols were mapped to human gene symbols using biomart as an R package. It is important to also note that one requirement of WGCNA with multiple models is that the same list of genes for each tissue sample should be used in the analysis, which may lead to missing the information on genes that are not represented in every sample. Briefly, raw counts and probe intensity data were pre-processed using the Limma package [63] in the R environment. Count data were transformed by mean-variance modelling at the observational level (voom) [64] before all studies were subjected to quantile normalization and data quality control as recommended for WGCNA. Finally, for differential gene expression analysis, raw count data were voom transformed and array data were log-transformed in the Limma package [63] in the R environment, followed by quantile normalization and data quality control.

### 4.3. Weighted Gene Correlation Analysis

Unsigned co-expression networks were built using the WGCNA 1.63 package in R software [65]. Clusters of genes that behaved similarly were grouped together into different color modules. These modules were related to specific traits. In heatmaps, red represents genes upregulated within that dataset and green represents genes downregulated within that dataset. The top 1000 connections within a gene network were determined by WGCNA. For the multiple array consensus analysis, WGCNA was performed on the individual datasets first, as suggested by Langfelder and Horvath’s tutorial [65], using “1 step function for network construction and detection of consensus modules”. The default WGCNA soft thresholding power β in which co-expression was raised was chosen to calculate the adjacency of each data set. The soft thresholding power β was used to allow us to compare each data set by approximate scale-free topology, thus compensating for scale differences between data sets.

### 4.4. Module Selection 

The correlation between module eigengenes and clinical traits was analyzed to identify modules of interest that were significantly associated with clinical traits. For the purpose of this study, we identified the modules that were significantly correlated with disease status in all three datasets. The correlation values were then displayed within a heatmap. Gene significance (GS) was defined as the correlation between gene expression and each trait. In addition, module membership (MM) was defined as the association between gene expression and each module eigengene. Subsequently, the correlations between GS and MM were examined to verify certain module–trait associations. The correlation analyses in this study were performed using Pearson correlation as described in the WGCNA package [65].

### 4.5. Module Enrichment 

The genes in each module of interest were extracted from the network and enrichment analysis was performed to further explore the functions of the respective modules. The R package ‘clusterProfiler’ was used to perform Kyoto Encyclopedia of Genes and Genomes (KEGG) [66,67] pathway enrichment analysis. A statistical *p* value of <0.05 was set as the significance threshold, and the enrichment results of KEGG pathways in each module of interest module were obtained.

### 4.6. Module Visualisation and Identification of Hub Genes 

The intramodular connectivity of genes in the corresponding modules of interest was measured using module eigengene-based connectivity (kME). The top 30 genes of each module of interest, which represent the central status in the module gene network, were selected to visualize the subordinate module using String software [68]. Subsequently, one key module was chosen that exhibited the highest levels of positive or negative correlation with RTT to search for hub genes for RTT in the modules. The top ten genes with the highest kME were selected as the hub genes in the corresponding module [65] and their gene significance (GS) for RTT (disease status) and intramodular connectivity kME were determined to confirm the reliability of these hub genes.

### 4.7. Differential Gene Expression

Differential gene expression analysis was performed on six patient-derived datasets from the MT study (GSE6955) and seven iPSC-derived neuronal samples from the OH study (GSE107399). Samples in the MT study were taken from the superior frontal gyrus of patients with RTT and age-matched controls. From the OH study, seven samples were utilized for analysis. Of these, four were RTT mutants (including two experimental controls) and three isogenic controls. The datasets were analyzed in R using the EdgeR package (R Bioconductor). Firstly, genes with low expression and a CPM value ≤ 1 were filtered. Then, the remaining counts were used to generate linear models and statistical analysis was conducted. To identify overlapping differentially expressed genes from the OH and MT datasets corresponding to the skyblue module, the log fold change was noted for genes that overlapped were identified. 

## 5. Conclusions

Through this meta-analysis and sub-analysis of datasets belonging to a mouse model, using post-mortem brain and iPSC-derived neurons to identify dysregulated genes that underpin the RTT pathology, a set of genes common to all models were identified. Some genes, such as *BDNF*, *ADCY9*, *ATRX*, and *CACNA1A*, have previously been linked to RTT, while others, including *CCT5*, *RPSP*, and *PLCG1*, are potential disease-modifying genes. Interestingly, a previous transcriptomic study using DGE only in RTT human samples did not cidentify some of the molecular network hub genes identified in the current study [69]. Validating *BDNF*, a known target of MeCP2, demonstrates the utility of this bioinformatic approach in identifying therapeutic genes targets. Further exploration of these known and novel disease-modifying genes may provide a better understanding of the molecular mechanisms of RTT and pave the way for the investigation of novel therapeutic candidates.

## Figures and Tables

**Figure 1 ijms-23-11125-f001:**
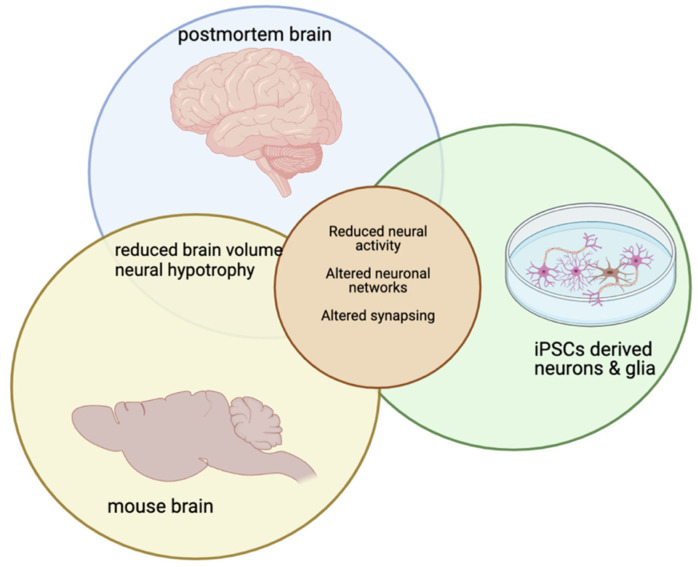
Cellular phenotype shared by mouse brain, post-mortem brain and iPSC-derived neurons and glia.

**Figure 2 ijms-23-11125-f002:**
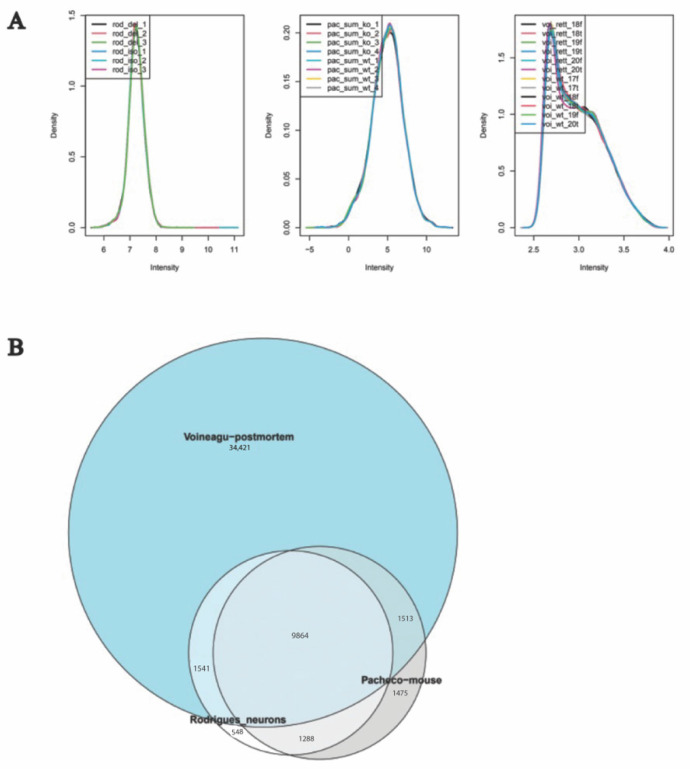
Data pre-processing. (**A**) Normalisation of datasets. Each panel represents a study in the following order (left to right): Rodrigues, Pacheco and Voineagu, each color line represents a sample. (**B**) Venn diagram showing genes in common between the three datasets.

**Figure 3 ijms-23-11125-f003:**
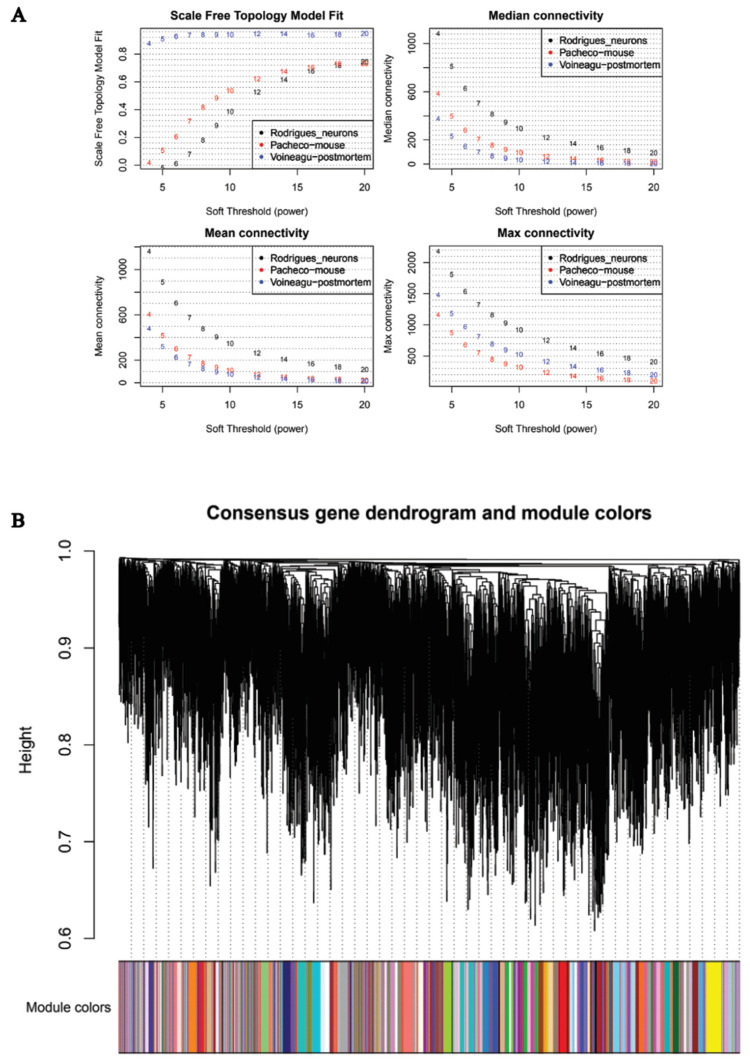
WGCNA analysis of the three datasets. (**A**) Analysis of network topology as a function of the soft-thresholding power for all three datasets. The panels show the scale-free fit index (top left), median connectivity (in degrees, top right), mean connectivity (in degrees, bottom left) and the maximum connectivity (in degrees, bottom right). (**B**) Clustering dendrogram of genes. Gene clustering tree (dendrogram) obtained by hierarchical clustering of adjacency-based dissimilarity. The coloured row below the dendrogram indicates module membership identified by the dynamic tree cut method and the assigned module colour.

**Figure 4 ijms-23-11125-f004:**
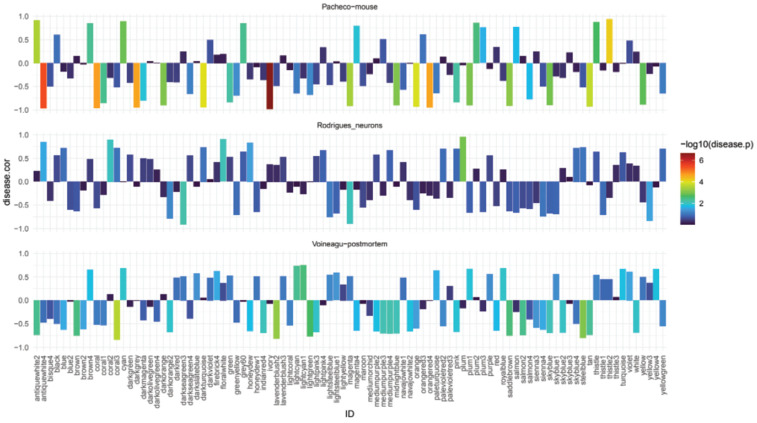
Module-feature associations. Each row corresponds to a module eigengene and its correlation with the clinical phenotype (disease status). Correlation coefficient is represented in log10 scale, where blue corresponds to a negative one correlation coefficient, and red corresponds to positive one correlation coefficient.

**Figure 5 ijms-23-11125-f005:**
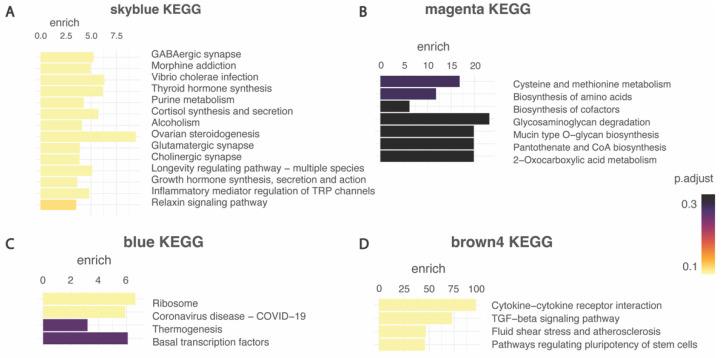
Enrichment analysis in interesting modules. (**A**) Skyblue module, (**B**) magenta module, (**C**) blue modules (**D**) brown4 module. Results include level of significance of each pathway enrichment using The Kyoto Encyclopedia for Genes and Genomes (KEGG) calculated and expressed in adjusted *p*-value, yellow represents more significant and purple least significant.

**Figure 6 ijms-23-11125-f006:**
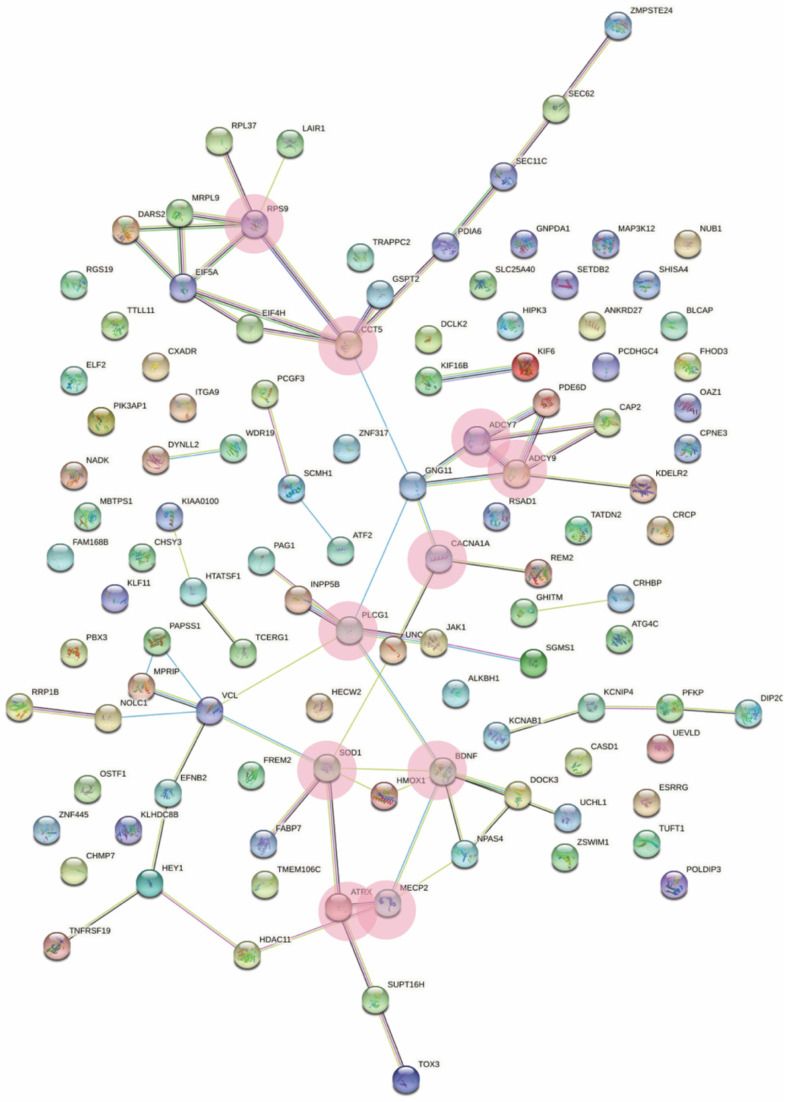
String diagram depicting the gene network of the skyblue module. Pink circles depict the identified hub genes.

**Figure 7 ijms-23-11125-f007:**
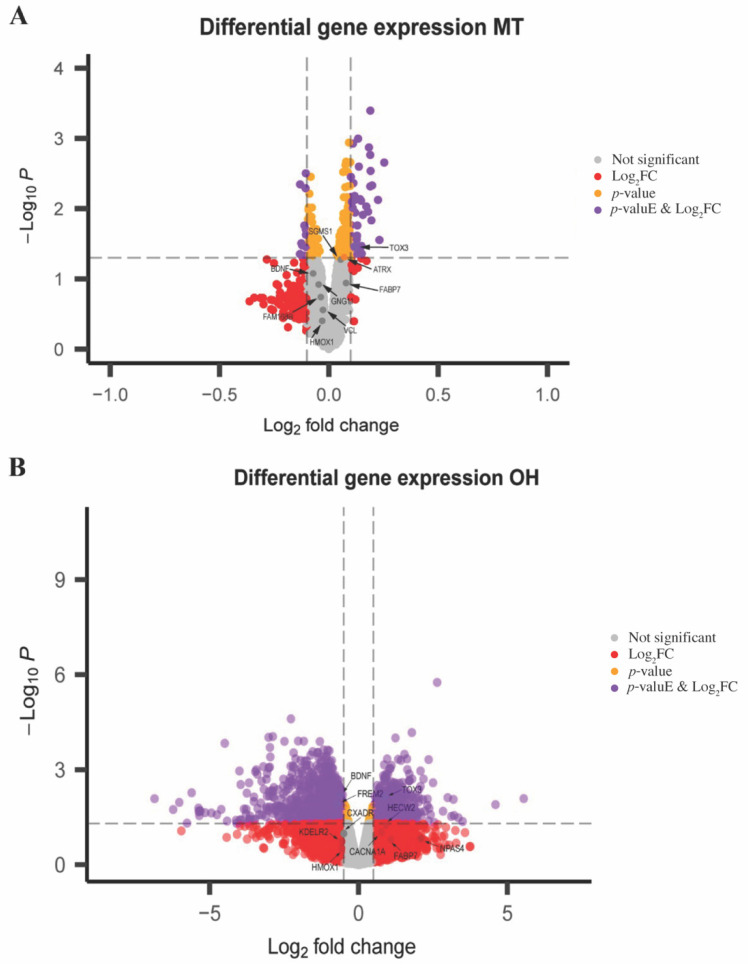
Volcano plot showing similarities in differentially expressed genes between the skyblue module and selected datasets. (**A**) Volcano plot of shared gene expression profiles from patients with RTT and age matched controls within skyblue and MT. The log_2_FC (*x*-axis) of each gene is plotted against the −log_10_*P* (*y*-axis). Expression difference is considered significant for log_2_FC of 0.1, and *p*-value < 0.05, as indicated by the purple-coloured points. The grey points represent genes with non-significant differences in expression. (**B**) Volcano plot of similarly differentially expressed genes in the skyblue module and OH dataset comparing patient-derived neuronal samples and controls. The log_2_FC (*x*-axis) of each gene is plotted against the −log_10_*P* (*y*-axis). Significant differences in gene expression, where log_2_FC is 0.5, and *p*-value < 0.05, are depicted by the purple points. The grey points represent genes with non-significant differences in expression.

**Figure 8 ijms-23-11125-f008:**
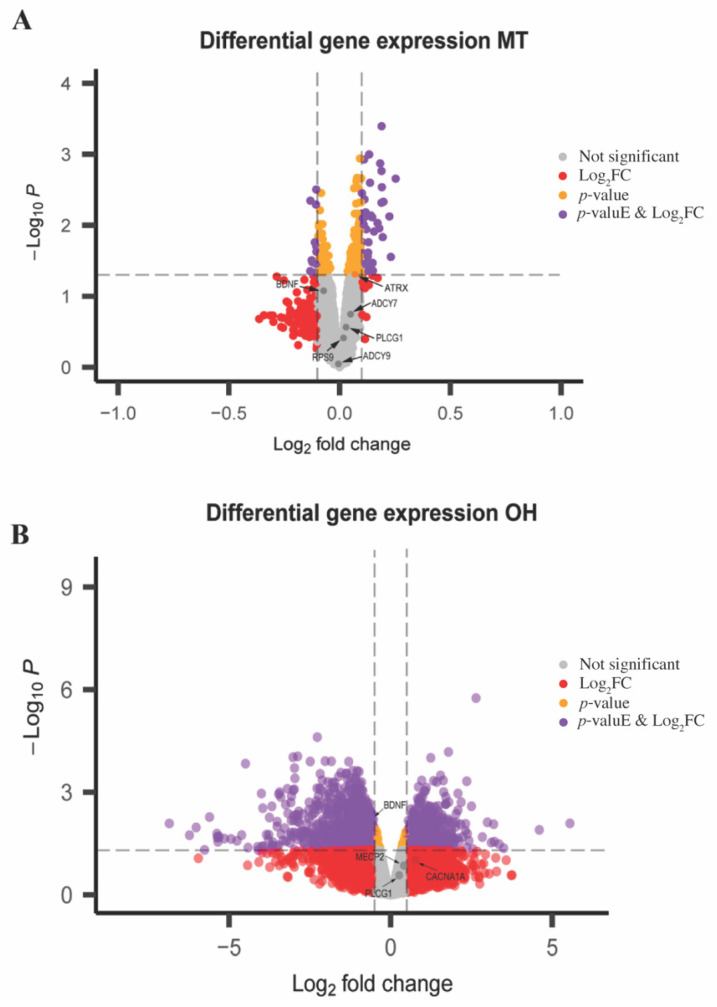
Expression of hub genes from skyblue module in MT and OH datasets. (**A**) Volcano plot of differential gene expression profiles in the MT data highlighting hub genes with significant changes. *ATRX*, *BDNF*, *ADCY7*, *PLCG1*, *RPS9* and *ADCY9* are depicted. (**B**) Volcano plot of differentially expressed genes in the OH dataset, highlighting the hub genes *BDNF*, *CACNA1A*, *MECP2* and *PLCG1*. Genes with significant changes in logFC (*p* < 0.05) are denoted by purple points. Genes with no significant changes in expression are represented by grey points.

**Table 1 ijms-23-11125-t001:** Summary of samples used in WGCNA. WT refers to wild type, RTT refers to Rett Syndrome and NA refers to not applicable; F refers to feminine and M to masculine.

Study	Sample	Age	Gender	Tissue	Disease State	Mutation
Post-mortem human brain GSE75303	GSM1949097	19y 231d	F	frontal cortex	WT	NA
	GSM1949098	17y 28d	F	frontal cortex	WT	NA
	GSM1949099	17y 28d	F	temporal cortex	WT	NA
	GSM1949100	20y 228d	F	temporal cortex	WT	NA
	GSM1949101	18y 138d	F	frontal cortex	WT	NA
	GSM1949102	18y 138d	F	temporal cortex	WT	NA
	GSM1949103	18y 130d	F	frontal cortex	RTT	c.378-2A > G
	GSM1949104	18y 130d	F	temporal cortex	RTT	c.378-2A > G
	GSM1949105	20y 356d	F	frontal cortex	RTT	c.763C > T
	GSM1949106	20y 356d	F	temporal cortex	RTT	c.763C > T
	GSM1949107	19y 280d	F	frontal cortex	RTT	c.451G > T
	GSM1949108	19y 280d	F	temporal cortex	RTT	c.451G > T
iPSC-derived neurons GSE123753	GSM3510829	NA	F	neurons	WT	isogenic
	GSM3510835	NA	F	neurons	MT	Exon 3–4 deletion
	GSM3510857	NA	F	neurons	WT	isogenic
	GSM3510863	NA	F	neurons	MT	Exon 3–4 deletion
	GSM3510877	NA	F	neurons	WT	isogenic
	GSM3510883	NA	F	neurons	MT	Exon 3–4 deletion
Mouse brain GSE96684	GSM2538276	P60	M	cortex	WT	NA
	GSM2538277	P60	M	cortex	WT	NA
	GSM2538278	P60	M	cortex	WT	NA
	GSM2538279	P60	M	cortex	WT	NA
	GSM2538280	P60	M	cortex	RTT	R168X
	GSM2538281	P60	M	cortex	RTT	R168X
	GSM2538282	P60	M	cortex	RTT	R168X
	GSM2538283	P60	M	cortex	RTT	R168X

**Table 2 ijms-23-11125-t002:** Hub genes and functions. Information adapted from GeneCards.

Gene Symbol	Gene Function
*MECP2*	Methyl-CpG-binding protein 2; a chromatin-associated protein that can both activate and repress transcription. It is required for maturation of neurons and is developmentally regulated.
*BDNF*	Brain-derived neurotrophic factor; during development, promotes the survival and differentiation of selected neuronal populations of the peripheral and central nervous systems. Participates in axonal growth, pathfinding and in the modulation of dendritic growth and morphology. Major regulator of synaptic transmission and plasticity at adult synapses in many regions of the central nervous system (CNS). The versatility of BDNF is emphasised by its contribution to a range of adaptive neuronal responses including long-term potentiation (LTP), long-term depression (LTD), certain forms of short-term synaptic plasticity.
*CCT5*	T-complex protein 1 subunit epsilon; a molecular chaperone that assists the folding of proteins upon ATP hydrolysis. As part of the BBS/CCT protein complex it may play a role in the assembly of BBSome, a complex involved in ciliogenesis, regulating transport vesicles to the cilia. Known to play a role in vitro in the folding of actin and tubulin.
*CACNA1A*	Voltage-dependent P/Q-type calcium channel subunit alpha-1A; voltage-sensitive calcium channels (VSCC) mediate the entry of calcium ions into excitable cells and are also involved in a variety of calcium-dependent processes, including muscle contraction, hormone or neurotransmitter release, gene expression, cell motility, cell division and cell death. The isoform alpha-1A gives rise to P and/or Q-type calcium currents. P/Q-type calcium channels belong to the ‘high-voltage activated’ (HVA) group and are blocked by the funnel toxin (Ftx) and by omega-agatoxin-IVA (omega-Aga-IVA).
*ADCY9*	Adenylate cyclase type 9; an adenylyl cyclase that catalyses the formation of the signalling molecule cAMP in response to activation of G-protein-coupled receptors. Contributes to signalling cascades activated by CRH (corticotropin-releasing factor), corticosteroids and beta-adrenergic receptors.
*ADCY7*	Adenylate cyclase type 7; a membrane-bound, calcium-inhibitable adenylyl cyclase.
*ATRX*	Transcriptional regulator ATRX; involved in transcriptional regulation and chromatin remodelling. Facilitates DNA replication in multiple cellular environments and is required for efficient replication of a subset of genomic loci. Binds to DNA tandem repeat sequences in both telomeres and euchromatin, and in vitro binds DNA quadruplex structures. May helpin stabilising G-rich regions into regular chromatin structures by remodelling G4 DNA and incorporating H3.3-containing nucleosomes. Catalytic component of the chromatin remodelling complex ATRX:DAXX, which has ATP-dependent DNA translocase activity.
*RPS9*	Small subunit ribosomal protein s9e; ribosomal protein S9.
*SOD1*	Superoxide dismutase [Cu-Zn]; destroys radicals that are normally produced within the cells and toxic to biological systems.
*PLCG1*	1-phosphatidylinositol 4,5-bisphosphate phosphodiesterase gamma-1; mediates the production of the second messenger molecules diacylglycerol (DAG) and inositol 1,4,5-trisphosphate (IP3). Plays an important role in the regulation of intracellular signalling cascades. Becomes activated in response to ligand-mediated activation of receptor-type tyrosine kinases, such as PDGFRA, PDGFRB, FGFR1, FGFR2, FGFR3 and FGFR4. Plays a role in actin reorganisation and cell migration.

**Table 3 ijms-23-11125-t003:** Summary of samples used in the DGE analysis. WT refers to wild type, RTT refers to Rett Syndrome, and NA refers to not applicable; F refers to feminine and M to masculine.

Study	Sample	Gender	Tissue	Disease State	Mutation
Post-mortem human brain (MT) GSE6955	GSM160306	F	Superior Frontal Gyrus	WT	NA
	GSM160307	F	Superior Frontal Gyrus	RTT	c.316C > T
	GSM160308	F	Superior Frontal Gyrus	WT	NA
	GSM160309	F	Superior Frontal Gyrus	RTT	c.316C > T
	GSM160310	F	Superior Frontal Gyrus	WT	NA
	GSM160311	F	Superior Frontal Gyrus x	WT	NA
	GSM2866278	F	neurons	WT	Isogenic (c.1461A > G)
	GSM2866279	F	neurons	WT	Isogenic (c.705delG)
iPSC-derived neurons (OH) GSE107399	GSM2866281	F	neurons	WT	Isogenic (c.705delG)
	GSM2866282	F	neurons	RTT	c.1461A > G (replicate 1)
	GSM3510883	F	neurons	RTT	c.1461A > G (repiclate 2)
	GSM3510885	F	neurons	RTT	c.705delG (replicate 1)
	GSM3510886	F	neurons	RTT	c.705delG (replicate 2)

## Data Availability

Code provided on request.

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
