# Peer review of "Meta-Analysis Identifies BDNF and Novel Common Genes Differently Altered in Cross-Species Models of Rett Syndrome"

_ijms, 2022, doi:10.3390/ijms231911125_

Round 1

Reviewer 1 Report

The study by Haase et al. performed a meta-analysis on Rett syndrome. The findings provide additional information/hypothesis for researchers to understand the disease mechanism. Since there are limited data set on this rare disease, it is reasonable/ acceptable to apply the described dataset (mixing with different species, in vitro in vivo studies etc). Basically, the data has merit to publish. I have the following concerns/ suggestions for improvement. Authors should check their works more carefully, and make sure it is understandable/ flow of the information.

Major:

I suggest the introduction could include more background information of the disease mechanism. There are several papers in other models demonstrated how the mecp2, for example , could affect the neural cell in zebrafish and other models.

Since the study included three different data set that have different features (L115-122), the authors should stay (describe) the limitations of using these data for comparison at the end of the discussion.

L132, what are the settings of “normalized and filtered”. More details should be given (not only the L133-135). How these settings apply to the three diff samples.

L226: define OH and MT. I cannot find them from the description. MT is defined in table 1. Please state in the main text.

L228 / 546 GSE 6955 or GSE107399 dataset, what are they? I only find these access number 2 times in the MS, and any background information for that?

L278, I notice the validation was done from “ iPSC-derived neurons from a male RTT patient and a paedritic control as well as a male post mortem forebrain sample and a control. So, only one RTT sample?, and the figure 9 shows n = 2. Nevertheless, I do not think that the data is representative or suitable to show in the main figure, please put it as sup. And L281, mentioned mecp2 and bdnf were excluded that I cannot agree, as that will be necessary to show/confirm you samples are RTT, especially when your n number is small….

Minor:

Orientation of figure 4.

Abstract: L29 “,” before human

Figure 8 is not clear, at least please highline or box the selected gene for easy reading.

Conclusion,

L610, “.” Before the Validating

L615. What is the ref [230]

L493. “s”Finally

There are numerous of typo and errors in the text, The authors should check these minor issue before submission.

Please double check the ref list, in the main text the ref has  19x - 2xx , but the ref you show has only 33 references.

Author Response

I suggest the introduction could include more background information of the disease mechanism. There are several papers in other models demonstrated how the mecp2, for example, could affect the neural cell in zebrafish and other models.

Since the study included three different data set that have different features (L115-122), the authors should stay (describe) the limitations of using these data for comparison at the end of the discussion.

Thank you very much for reviewing this article and for all your valuable comments/suggestions. We have now included a paragraph in the introduction addressing the disease mechanism and the effects of MECP2 mutations at a cellular level (L75-74). We have also included an explanation on the limitation of the use of these models and the bioinformatic approach used (L922-925).

L132, what are the settings of “normalized and filtered”. More details should be given (not only the L133-135). How these settings apply to the three diff samples.

We have included specific information describing how each dataset was normalised in the methods section 4.2.

 L226: define OH and MT. I cannot find them from the description. MT is defined in table 1. Please state in the main text.

We have amended this as MT and mutant may be confusing, we have now changed MT to RTT.

L228 / 546 GSE 6955 or GSE107399 dataset, what are they? I only find these access number 2 times in the MS, and any background information for that?

This section has been amended. Furthermore, we have included Table 3 including all samples used in these two studies. 

L278, I notice the validation was done from “ iPSC-derived neurons from a male RTT patient and a paedritic control as well as a male post mortem forebrain sample and a control. So, only one RTT sample?, and the figure 9 shows n = 2. Nevertheless, I do not think that the data is representative or suitable to show in the main figure, please put it as sup. And L281, mentioned mecp2 and bdnf were excluded that I cannot agree, as that will be necessary to show/confirm you samples are RTT, especially when your n number is small….

Upon reconsideration and agreeing with reviewer 1 and 2 comments we have decided to not include this data as the sample size is small and it may not contribute to the in silico findings.

Minor:

Orientation of figure 4.

This has been amended

Abstract: L29 “,” before human

Figure 8 is not clear, at least please highline or box the selected gene for easy reading.

We have re-done Figures 7 and 8 and highlighted the important genes as mentioned in the legend of both figures.

Conclusion,

L610, “.” Before the Validating

This has been amended

L615. What is the ref [230]

All references have been amended.

L493. “s”Finally

This has been amended

There are numerous of typo and errors in the text, The authors should check these minor issue before submission.

These have been corrected

Please double check the ref list, in the main text the ref has  19x - 2xx , but the ref you show has only 33 references.

Thank you for this comment, references have now been amended. 

Reviewer 2 Report

The authors conducted a meta-analysis to uncover common altered genes in humans, as well as in vitro and mouse models of RETT syndromes. I believe that the authors provide some nice and important findings.

While I believe that most of the bioinformatic analyses are quite solid, as bioinformatics is not one of my strengths, the results reported in figure 9 seem to be very preliminary and the weak part of this work, and in the present form cannot support the authors conclusions. However, it would be a great addition, and the authors should consider strengthening it.

The introduction is very well written and the rationale behind this study is well explained and supported. 

Major Concerns:

Figure 9: The number of independent samples n=2 is too low, that could be understandable for the postmortem samples, but it is not acceptable for the in vitro part. The number of independent samples should be increased to at least 5 before any information can be extrapolated or these can confirm and support the authors findings, taking into account the variability in cell transfection etc…..

This experiment is missing a control (normal postmortem human brain tissue), the expression levels in human post-mortem samples should not be ‘normalised’ to the “control cell lines”. 

Likely the signals/values, shown in figure 9, were obtained by first normalising to GAPDH and then to the values of the control cells. Hence, are they reported as ratio to control cells?

Figure legend to fig 9:  I am not sure to what cell lines the authors are referring in the fig legend, since in the methods they wrote that the neurones were generated from a male RTT iPSc and a normal paediatric patient, or were these immortalised cells purchased from a company? Please describe.

Please explain the rational about using cells from a male patient? There is no demographical information on this subject, the normal patient, or the post-mortem brain samples used for RNA extraction and purification. Were those males or females, age, cause of death? How were these samples acquired?

In the authors’ contribution paragraph, there is no description of who has performed these experiments. Paragraphs 4.8 and 4.9 are not carefully described and are devoid of important details, they should be written with plenty of details to allow for a proper assessment of the results.

The title of paragraph 4.8 is misleading as it mostly describes the preparation of the lentiviruses. However, at the beginning of this paragraph, the authors mention that to generate excitatory neurons they used a method previously reported (ref 32). Is this the same or similar to the one described in 4.9? Perhaps, these two paragraphs do need some reorganisation.

There is no description of how the RT- qPCR itself was performed, what machine was used, reagents, mastermix for both RT and qPCR?   Was it one or two steps? What did the authors use as standards?

How was the RNA extracted from the post-mortem brains?
How was the RNA purity and quality determined? how was the RNA quantified? 

What was the starting concentration of RNA used in the RT reactions?

“cDNA extracted from the iNeurons” this is an incorrect statement as the tRNA or mRNA is extracted/purified from tissue or cell and then reverse transcribed ……..

What housekeeping gene(s) was(were) used? These should be reported in the methods and not in the figure legends.

How were the signals normalised and quantified? 

Language: 

The authors should consider explaining some of the terminology used in the manuscript in consideration of the broad readership of this journal, many are not familiar with all of them.

Data is the plural of datum, therefore the verb after should be “were”. Please revise throughout the manuscript and consider keeping the verbs consistent.

For instance, in the following sentence: “The data WERE normalised and filtered prior to WGCNA analysis (Figure 2). “ the authors actually do normalise ALL of the data derived from the three studies. (I do understand that ‘data’ is often used as a singular noun and is considered correct in English.)

Suggestion: any missing DATA count was eliminated. 

Paedritic=paediatric

Author Response

The authors conducted a meta-analysis to uncover common altered genes in humans, as well as in vitro and mouse models of RETT syndromes. I believe that the authors provide some nice and important findings.

While I believe that most of the bioinformatic analyses are quite solid, as bioinformatics is not one of my strengths, the results reported in figure 9 seem to be very preliminary and the weak part of this work, and in the present form cannot support the authors conclusions. However, it would be a great addition, and the authors should consider strengthening it.

The introduction is very well written and the rationale behind this study is well explained and supported.

Major Concerns:

Figure 9: The number of independent samples n=2 is too low, that could be understandable for the postmortem samples, but it is not acceptable for the in vitro part. The number of independent samples should be increased to at least 5 before any information can be extrapolated or these can confirm and support the authors findings, taking into account the variability in cell transfection etc.....

This experiment is missing a control (normal postmortem human brain tissue), the expression levels in human post-mortem samples should not be ‘normalised’ to the “control cell lines”.

Likely the signals/values, shown in figure 9, were obtained by first normalising to GAPDH and then to the values of the control cells. Hence, are they reported as ratio to control cells?

Figure legend to fig 9: I am not sure to what cell lines the authors are referring in the fig legend, since in the methods they wrote that the neurones were generated from a male RTT iPSc and a normal paediatric patient, or were these immortalised cells purchased from a company? Please describe.

Please explain the rational about using cells from a male patient? There is no demographical information on this subject, the normal patient, or the post-mortem brain samples used for RNA extraction and purification. Were those males or females, age, cause of death? How were these samples acquired?

In the authors’ contribution paragraph, there is no description of who has performed these experiments. Paragraphs 4.8 and 4.9 are not carefully described and are devoid of important details, they should be written with plenty of details to allow for a proper assessment of the results.

The title of paragraph 4.8 is misleading as it mostly describes the preparation of the lentiviruses. However, at the beginning of this paragraph, the authors mention that to

1

generate excitatory neurons they used a method previously reported (ref 32). Is this the same or similar to the one described in 4.9? Perhaps, these two paragraphs do need some reorganisation.

There is no description of how the RT- qPCR itself was performed, what machine was used, reagents, mastermix for both RT and qPCR? Was it one or two steps? What did the authors use as standards?
How was the RNA extracted from the post-mortem brains?
How was the RNA purity and quality determined? how was the RNA quantified?
What was the starting concentration of RNA used in the RT reactions?
“cDNA extracted from the iNeurons” this is an incorrect statement as the tRNA or mRNA is extracted/purified from tissue or cell and then reverse transcribed ........
What housekeeping gene(s) was(were) used? These should be reported in the methods and not in the figure legends.
How were the signals normalised and quantified?

Thank you very much for reviewing the present manuscript and for all your comments/suggestions. Upon further consideration amongst the authors and having taken into account comments made by all three reviewers, we have decided not to include the RT-qPCR hub gene validation. We have plans for further characterising the identified hub genes in future projects by the inclusion of more patient derived cells, however at this time we consider that the results obtained through the bioinformatic analysis will pave the way for further studies.

Language:

The authors should consider explaining some of the terminology used in the manuscript in consideration of the broad readership of this journal, many are not familiar with all of them.

Thank very much for this observation, we have now included further explanations in terms of data handling including normalisation methods along the manuscript for readers who may not be familiar with this approach.

Data is the plural of datum, therefore the verb after should be “were”. Please revise throughout the manuscript and consider keeping the verbs consistent.

For instance, in the following sentence: “The data WERE normalised and filtered prior to WGCNA analysis (Figure 2). “ the authors actually do normalise ALL of the data derived from the three studies. (I do understand that ‘data’ is often used as a singular noun and is considered correct in English.)

Suggestion: any missing DATA count was eliminated. Paedritic=paediatric

Thank you for this comment, this has now been amended.

Reviewer 3 Report

The authors performed  WGCEA analysis and tried to identify commonly differentially altered genes of RTT . The autors identified modules correlating clinical traits, and predicted common pathways related to RTT. This approach enables to discover intracellular pathways related to RTT and to find therapeutic targets. In general, the results obtained from in silico analysis should be evaluated by comparison with in vivo evidence such as experimental results and clinical traits. However, in this article, this evaluation is not clear and not solid, because of inadequate data interpretation and immature experimental results. 

I picked up several points below, which should be reasonably explained by showing reliable experimental results.

Major points:

1. "2.4 Different Brain Tissues/Models Generate Modules of Dysregulated Genes" and Figuere 4.: The auhtors "found four models to be significantly dysregulated across all three datasets: brown4, blue, magenta and skyblue". By my understanding, "dysregulated across all three database" means in all three datasets, "disease cor."  shows negative value. Judged from Figure 4., Skyblue dataset fulfill this, however, brown4, blue, and magenta did not. If three (brown4, blue and magenta) did not dysregulated gene dataset, "2.5 Module Analysis" must be replaced (for example coral, saddle brown etc.). Please explain this.

2. Figure 7. is confusing, in main text and figure legend, explanation of panel A is for MT dataset and panel B for OH dataset. However, panel A in the figure is for OH, and panel B for MT. Please check again and present correct figure-legend-text set.

3. IFigure 8. has same problem as in Fig.7. In main text and figure legend, explanation of panel A is for MT dataset and panel B for OH dataset. However, panel A in the figure is for OH, and panel B for MT. Please check again and present correct figure-legend-text set.

4. In "2.8 Hub gene PCR validation" and Figure 9, qPCR data presented is not raliable, because N number is only 2 and statistical significance was not shown. Please increase N (at least 3, larger is better), calculate and show statistical significance. In bar graph, plot original data as dots. Please present clearly that the gene expression levels are up-regulated  or down-regulated showing calculated expression level values. Then discuss again the relationship between mata-analysis results and real expression pattern based on novel reliable data. 

Minor points: 

1. In Figure 3., line 163, please show "merged module colours". Line 157, (a) to (A). Line 160, (b) to (B).

2. In figure 4., -log10(disease.P) should be -log10(disease.P).

3. In Figure 5., line 190, (a), (b), (c) to (A), (B), (C), respectively.

4. In Figure 7., (a), (b) ro (A), (B).

5. In Figure 8., (a), (b) ro (A), (B).

6. In Figure 9. In the graph, as the bar for ADCY7 iNeurons is missing, please present this.

7. In 4.7 Differential gene expression, lane 546, "six iPS-derived neurons" is true? Not six but eight?

8. In 4.9 Culture and RNA extraction, line 601, "cDNA extracted from" should be "cDNA synthesized using RNA extracted from the iNeurons...".

9. Line 93, "Error...".

10. Line 63, "postpost-morten".

11. Line171, "0.0.5".

12. Line 185, "Error...".

Author Response

  1. "2.4 Different Brain Tissues/Models Generate Modules of Dysregulated Genes" and Figuere 4.: The auhtors "found four models to be significantly dysregulated across all three datasets: brown4, blue, magenta and skyblue". By my understanding, "dysregulated across all three database" means in all three datasets, "disease cor."  shows negative value. Judged from Figure 4., Skyblue dataset fulfill this, however, brown4, blue, and magenta did not. If three (brown4, blue and magenta) did not dysregulated gene dataset, "2.5 Module Analysis" must be replaced (for example coral, saddle brown etc.). Please explain this.

Thank you very much for reviewing the present article and for all your helpful comments/suggestions. We have reworded this paragraph to reduce any confusion between correlation and significance. There are more than four modules correlated to disease status however the correlation was only significant for the four mentioned in paragraph 2.4.

  1. Figure 7. is confusing, in main text and figure legend, explanation of panel A is for MT dataset and panel B for OH dataset. However, panel A in the figure is for OH, and panel B for MT. Please check again and present correct figure-legend-text set.

We thank you for this, both figures have been replaced as well as their corresponding legends. Furthermore, we have included a table listing the samples included in both studies (Table 3)

  1. IFigure 8. has same problem as in Fig.7. In main text and figure legend, explanation of panel A is for MT dataset and panel B for OH dataset. However, panel A in the figure is for OH, and panel B for MT. Please check again and present correct figure-legend-text set.

This has been corrected.

  1. In "2.8 Hub gene PCR validation" and Figure 9, qPCR data presented is not raliable, because N number is only 2 and statistical significance was not shown. Please increase N (at least 3, larger is better), calculate and show statistical significance. In bar graph, plot original data as dots. Please present clearly that the gene expression levels are up-regulated or down-regulated showing calculated expression level values. Then discuss again the relationship between mata-analysis results and real expression pattern based on novel reliable data. 

Thanks for your observation, after further consideration, we have decided to remove this section from the present study and focus only on the in-silico study.

Minor points: 

  1. In Figure 3., line 163, please show "merged module colours". Line 157, (a) to (A). Line 160, (b) to (B).
  2. In figure 4., -log10(disease.P) should be -log10(disease.P).
  3. In Figure 5., line 190, (a), (b), (c) to (A), (B), (C), respectively.
  4. In Figure 7., (a), (b) ro (A), (B).
  5. In Figure 8., (a), (b) ro (A), (B).

Point 1-5 have been addressed this in the manuscript.

  1. In Figure 9. In the graph, as the bar for ADCY7 iNeurons is missing, please present this.

Figure 9 and related text has subsequently been removed.

  1. In 4.7 Differential gene expression, lane 546, "six iPS-derived neurons" is true? Not six but eight?

Thanks for this question, there were some consistency problem throughout the manuscript in terms of how we referred to this datasets samples. We have now made the number consistent throughout the manuscript. There were seven samples total, four of those were RTT and of those two were experimental replicates.

  1. In 4.9 Culture and RNA extraction, line 601, "cDNA extracted from" should be "cDNA synthesized using RNA extracted from the iNeurons...".

This section has now been removed.

  1. Line 93, "Error...".
  2. Line 63, "postpost-morten".
  3. Line171, "0.0.5".
  4. Line 185, "Error...".

Thank you for this, we have addressed all these comments in text.

Round 2

Reviewer 1 Report

As mentioned in my last comment, i suggest the authors to add more background of the rett by summarizing the findings not only in mammalian but the other model like zebrafish, authors could find several papers when searching the pubmed, that would add merits to this paper for the rare disease.

L541 onwards, authors have stated the limitation of the study, which is essential to remind the readers on such shortness. However, authors try to "re-amplify" the merits of combining the two "methods" that I do not agree. The "so-called" validation is not well-concluded, or on the other hand, as shown in their deleted figure. I do not agree with the conclusion that 'support the power of this approaches ...."

The reason that I agree of combining multiple models in this study is due to the limited occurance and database of such disease; that the data could somehow provide certain information to the field. I request the authors to clearly state out the limitation, but not to mention the "successfulness" of the methods that I do not totally agree.  

Author Response

As mentioned in my last comment, i suggest the authors to add more background of the rett by summarizing the findings not only in mammalian but the other model like zebrafish, authors could find several papers when searching the pubmed, that would add merits to this paper for the rare disease.

We apologise for not addressing this question appropriately. We have now included the following information :

'At a cellular level, MeCP2 expression is critical for neuronal maturation and neuronal function. In hemizygous male patients, loss-of-function mutations in MECP2 cause neonatal encephalopathy, which is usually fatal before the age of two, and in heterozygous female patients it results in a severe neurological phenotype [2]. MeCP2 is a pleiotropic protein mediating early events of neurodevelopment, including neurogenesis, migration, and patterning, where it predominantly acts as a methyl-DNA binding protein and controls transcriptional regulation. The protein is highly conserved among mammals showing 95% protein sequence identity between human and mouse, and to a lesser extent, between humans and zebrafish (48%). However, the genomic structure and expression patterns of MeCP2 in zebrafish and mammals are similar, suggesting probable conserved functions [3]). MeCP2 contains two highly evolutionarily conserved domains, the methyl-CpG-binding domain (MBD) and the transcriptional repression domain (TRD) and acts as a transcriptional repressor an activator. Upon binding to CpG islands, MeCP2 forms an inhibitory transcription complex through interactions of the transcriptional repression domain with cofactors including Sin3A and histone deacetylase 1. MeCP2 also acts as a transcription activator to regulate gene expression by either long-range chromatin remodeling or by regulating RNA splicing ([4, 5]. MeCP2-deficiency leads to an excitation/inhibition (E/I) imbalance in the brain and is recognised as the leading cellular and synaptic hallmark of the disorder resulting in stereotypic hand movements, impaired motor coordination, breathing irregularities, seizures, and learning/memory dysfunctions [6].'

L541 onwards, authors have stated the limitation of the study, which is essential to remind the readers on such shortness. However, authors try to "re-amplify" the merits of combining the two "methods" that I do not agree. The "so-called" validation is not well-concluded, or on the other hand, as shown in their deleted figure. I do not agree with the conclusion that 'support the power of this approaches ...."

We have removed the hyperbole about our analytical approach and focused on the identification of BDNF, known RTT target gene, to round up the discussion in a positive note.

The reason that I agree of combining multiple models in this study is due to the limited occurance and database of such disease; that the data could somehow provide certain information to the field. I request the authors to clearly state out the limitation, but not to mention the "successfulness" of the methods that I do not totally agree.  

We have moved the comment about the limitation of WGCNA to methodology section 4.2 and we have added a sentence remarking the use of data pre-processing to mitigate this limitation. However, with all due respect to the reviewer, there is not a ‘limited occurrence and database of such disease’. The limitations in understanding the underlying pathophysiology of RTT rather lie in the complexity of the disorder and therefore new methods in which to unravel the complexity of the disorder, such as we are proposing, are required. As BDNF is consistently associated with MECP2-deficiency in the literature, and our analysis identified BNDF as a consistently downregulated gene across all models and datasets, we are highlighting the success of this method as it consistently validates previous findings.  For such a rare disorder, this is very important as it suggests that our methodology of using multiple models could potentially be a successful model of identifying other commonly dysregulated genes.

Reviewer 2 Report

I have no further comments.

Author Response

Thanks for reviewing our article. 

Reviewer 3 Report

The authors had revised the manuscript following my suggestions and greatly improved. However, there remains several points.

Major points

1. To may previous question 1, the authors replied "Thank you very much for reviewing the present article and for all your helpful comments/suggestions. We have reworded this paragraph to reduce any confusion between correlation and significance. There are more than four modules correlated to disease status however the correlation was only significant for the four mentioned in paragraph 2.4.". 

However, nothing was reworded in this paragraph. Please reword this paragraph to reduce confusion.

2. In the Table 3 newly added in this version, for GSM160307 sample, the disease state was RTT, however the mutation state was "NA". Is this true?

3. In the Table 3, for GSM2866278, GSM2866279, and GSM2866281, the mutation states were Isogenic (c.1461A>G), Isogenic (c.705delG), and Isogenic (c.705delG). Is this true? If not, please correct. These are critical.

Minor points: 

1. In the Figure 7 and 8, differentially expressed genes were indicated by arrows, however gene names were too small and thin, better to enlearge and choose thick/bold font. 

2. line 634: please correct "24tilized"

3. line 638: please complete "eBayes()"

4. line 639: please complete "topTable()"

5. line 641: please complete "VLOOKUP()"

Author Response

Major points

1. To may previous question 1, the authors replied "Thank you very much for reviewing the present article and for all your helpful comments/suggestions. We have reworded this paragraph to reduce any confusion between correlation and significance. There are more than four modules correlated to disease status however the correlation was only significant for the four mentioned in paragraph 2.4.". 

However, nothing was reworded in this paragraph. Please reword this paragraph to reduce confusion.

Apologies for this, the revised and updated version of the manuscript contains this changes in L229-233

2. In the Table 3 newly added in this version, for GSM160307 sample, the disease state was RTT, however the mutation state was "NA". Is this true?

Thanks for this observation, we have corrected the mutation

3. In the Table 3, for GSM2866278, GSM2866279, and GSM2866281, the mutation states were Isogenic (c.1461A>G), Isogenic (c.705delG), and Isogenic (c.705delG). Is this true? If not, please correct. These are critical.

Yes, this information is correct. 

Minor points: 

  1. In the Figure 7 and 8, differentially expressed genes were indicated by arrows, however gene names were too small and thin, better to enlearge and choose thick/bold font.

We have now amended this figures. 

2. line 634: please correct "24tilized"

3. line 638: please complete "eBayes()"

4. line 639: please complete "topTable()"

5. line 641: please complete "VLOOKUP()"

We have decided to leave the syntax out of the paragraph as it is too long and disrupts the flow of the text and as per stated in the submission portal, we will provide the code upon request. 

Thanks again for reviewing this article.